# Assessing Forage Potential of the Global Collection of Finger Millet (*Eleusine coracana* (L.) Gaertn.) Germplasm Conserved at the ICRISAT Genebank

Chinnadurai Backiyalakshmi [1,2], Chakrapani Babu [1], Dagunapur Naresh Reddy [2], Varijakshapanicker Padmakumar [3], Kodukula V. S. V. Prasad [3], Vania Cristina Renno Azevedo [2] and Mani Vetriventhan [2,*]

1 Centre for Plant Breeding and Genetics, Tamil Nadu Agricultural University (TNAU), Coimbatore 641003, Tamil Nadu, India; backiyachinna93@gmail.com (C.B.); babutnau@gmail.com (C.B.)
2 International Crops Research Institute for the Semi-Arid Tropics (ICRISAT), Patancheru 502324, Telangana State, India; D.Naresh@cgiar.org (D.N.R.); v.azevedo@cgiar.org (V.C.R.A.)
3 International Livestock Research Institute (ILRI), ICRISAT Campus, Patancheru 502324, Telangana State, India; V.Padmakumar@cgiar.org (V.P.); K.V.PRASAD@cgiar.org (K.V.S.V.P.)
* Correspondence: m.vetriventhan@cgiar.org

**Abstract:** Finger millet is an important drought-tolerant and grain-nutrient dense food crop grown in semi-arid regions in Asia and Africa. The forage is used as a source of dry roughage for feeding livestock. In this study, the finger millet diversity panel (310 accessions and four controls) representing the global collection of the finger millet germplasm conserved at the ICRISAT genebank was assessed for forage quality and diversity in the years 2018 and 2019. Results of the study suggested that finger millet can generate stover yield ranging from 2890 to 10,779 kg ha$^{-1}$. Finger millet forage contained 6.47% to 8.15% of crude protein, >90% of dry matter content, 11.47% to 14.17% of ash content, 62.36% to 67.73% of neutral detergent fiber (NDF), 33.07% to 38.37% of acid detergent fiber (ADF), 3.95% to 4.80% of acid detergent lignin (ADL), 6.18% to 6.89% of metabolizable energy (ME) and 45.21% to 49.09% of in vitro organic matter digestibility (IVOMD) with the grain yield of 810 to 3698 kg ha$^{-1}$ at maturity stage. The per se performance between the races, regions, and biological status has been performed and differed significantly for important traits. The 314 accessions were grouped into five clusters based on the performance for food-feed traits. Except for crude protein, there was a positive relationship between forage quality-positive traits and grain yield, indicating that agronomic and forage quality traits could be improved simultaneously. The top 10 promising accessions for important forage quality traits and accessions with multiple forage quality traits were identified. This study provides a detailed understanding of the variability that exists in forage quality traits in crop residues and their association with agronomic traits in the finger millet germplasm. The identified top-performing accessions would be the key genetic resources for developing dual-purpose cultivars and the information from this study will be useful for future finger millet food-feed trait improvement.

**Keywords:** forage quality; food-feed trait; characterization; variability; genetic diversity

## 1. Introduction

Finger millet (*Eleusine coracana* (L.) Gaertn.), is a drought-tolerant cereal crop, well adapted to the adverse environment of the semi-arid tropics of Asia and Africa. The finger millet crop is highly self-pollinated and allotetraploid in nature (AABB) with the chromosome number 2n = 4x = 36. It has superior grain nutritional quality with multiple health benefits [1–6]. It can produce substantial forage and grain yield even under low input conditions [7]. In the context of emerging climate change, there is a need to tap the crop's potential for a dual purpose like other millets because of the increasing demand [8].

Finger millet is primarily grown as a food crop, but it can also be used as a forage crop, producing excellent hay and green forage for cattle, sheep, and goats [1,9–12]. In India, USA, Africa, and Ireland, finger millet straw is used as forage, but it is also used for other purposes in Africa, such as string and thatching [1,13]. In comparison to forage maize and sorghum, finger millet forage is high in micronutrients like calcium, phosphorus, and potassium [14] and contains 61% of total digestible nutrients [15], 105 to 156 g kg$^{-1}$ crude protein, 598 to 734 g kg$^{-1}$ neutral detergent fibre, 268 to 382 g kg$^{-1}$ acid detergent fibre, 597 to 730 g kg$^{-1}$ in vitro digestibility, and 387 to 552 g kg$^{-1}$ neutral detergent fibre digestibility [11]. Apart from yield and grain nutritional enhancement, improvement of the forage quality of the crop residues becomes an integral part of plant breeding [16,17]. Multi-dimensional crop improvement is needed to obtain a sustainable increase in food-forage traits [18]. In recent times, several studies have been reported in annual grasses towards the characterization, evaluation, and the market value for forage yield and feed quality in rice [17,19,20], wheat [21–24], maize [25], pearl millet [26], and sorghum [27,28]. In semi-arid regions, sorghum and pearl millet are the predominant cereal crops and it has the additional value of the crop-by product that can feed livestock without compromising the grain yield. To improve the forage quality of crop residues without comprising on grain yield, crop improvement work has focused to breed dual-purpose cultivars in different crops [28]. However, in the case of finger millet crop, forage improvement has not yet attracted the plant breeding community and only a few studies have been reported on forage quality characterization [11,13,29]. Trait variability is the most important criterion in the selection process, and it can be explored through the characterization of germplasm, which is the base material for any crop improvement program. With this background, we performed this study (i) to assess the finger millet germplasm crop residues for its forage quality; (ii) to study the association of forage quality traits with the agronomic traits; and (iii) to identify the promising finger millet germplasm for superior forage quality traits.

## 2. Materials and Methods

### 2.1. Germplasm Details

A diversity panel of 310 accessions, along with four control cultivars (GPU 26, MR 6, KMR 204, and VL 149) of finger millet, originating from 23 different countries, representing the four geographical regions in the world: Africa (160), Asia (136), Europe (6), North America (3) and nine Unknown origin accessions, were used for this study (Table S1). The panel represents all the races of finger millet: *vulgaris* (202 accessions), *plana* (48 accessions), *elongata* (31 accessions), *compacta* (28 accessions), and a few unclassified (5 accessions), and their subraces. The diversity panel consists of both landraces (264 accessions) and breeding material (50 accessions) (Table S1).

### 2.2. Experimental Details

The experiment was conducted in an alpha lattice design with three replications for two consecutive years (2018 and 2019 rainy seasons) in alfisols at ICRISAT, Patancheru, Telangana, India (17.53° N latitude, 78.27° E longitude, and 545 MSL) to evaluate the accessions for agronomic and forage quality traits. Sowing was done in the third week of July in both years. Year-wise weather parameters for the years 2018 and 2019 during the crop growth period (July–December) are provided in Figure 1. Each accession was planted in a single row of 4 m length with about 10 cm interplant spacing and row-to-row spacing of 60 cm. Crop-specific agronomic and plant protection measures were followed. Diammonium phosphate was applied at 100 kg ha$^{-1}$ as a basal dose and urea was applied at the rate of 100 kg ha$^{-1}$ as a topdressing at the interval of 35 days after sowing (DAS).

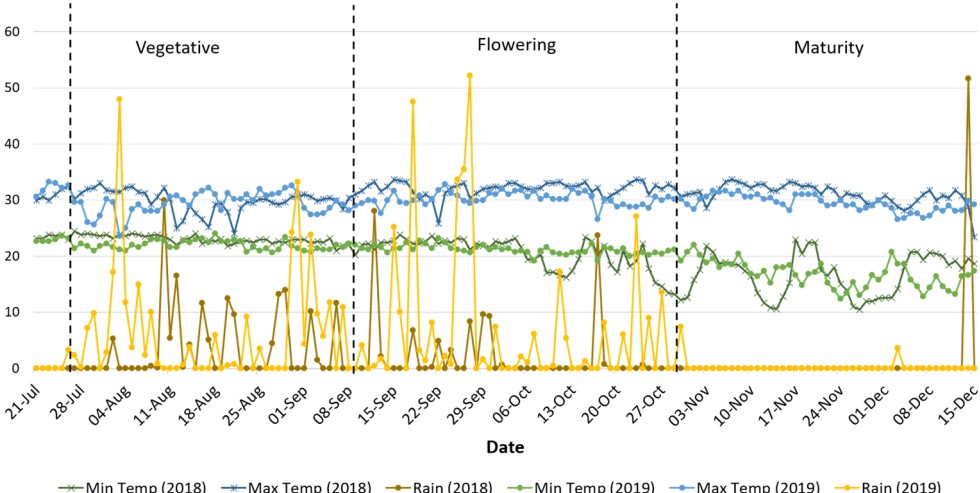

**Figure 1.** Weather parameter data of 2018 and 2019 rainy seasons during the finger millet crop growth stages.

### 2.3. Phenotyping for Agronomic Traits

Data on seven agronomic traits, namely days to 50% flowering, days to maturity, plant height, basal tillers number, grain yield, stover yield (dry), and harvest index (HI = grain yield/biological yield) were recorded and compared for the response on forage quality traits. Days to 50% flowering, days to maturity, grain yield, and stover yield (dry) were recorded on a plot basis. The remaining traits were recorded on five representative plants from each row of individual accessions. Grain and stover yield per plot were converted into yield kg ha$^{-1}$.

### 2.4. Forage Quality Trait Assessment

For forage quality traits analysis, 1 m length of plants in the middle of a 4 m row of each accession was harvested at maturity stage, sun-dried, chopped into small pieces, and ground into a powder to pass through a 1-mm sieve using hammer mills from the International Livestock Research Institute (ILRI) located at ICRISAT, Hyderabad. Then, the samples were sent to ILRI livestock nutritional laboratory, ICRISAT, Hyderabad, for forage quality traits assessment. The forage quality traits such as dry matter content (Dm, %), ash content (%), crude protein (CP, %), neutral detergent fiber (NDF, %), acid detergent fiber (ADF, %), acid detergent lignin (ADL, %), in vitro organic matter digestibility (IVOMD, %) and metabolizable energy (ME, MJ/kg Dm) were measured by near-infrared spectroscopy (NIRS) calibrated against conventional wet laboratory analyses [19,30]. The NIRS instrument was a FOSS Forage Analyzer 5000 (FOSS XDS RCA, Win ISI IV, Denmark) installed with the software package Win ISI II. The predicted values of all traits were corrected by dry matter (Dm), and the percentage values were used for statistical analyses. The $R^2$ values for global calibration of forage quality traits varied from 0.84 to 0.97 (0.84 for dry matter, 0.89 for ADL, 0.92 for IVOMD, 0.93 for ADF and ME, 0.96 for Ash and NDF, and 0.97 for CP).

### 2.5. Statistical Analysis

The significance between the accessions and the years was determined by estimated variance components in the linear mixed model for individual years and pooled data of two years using the restricted maximum likelihood (ReML) procedure in Genstat 20th edition (http://www.genstat.co.uk accessed on 7 January 2021), considering the year as fixed and treatment, block, replication as a random factor. The pooled mean data had been used for downstream analysis. The broad-sense heritability (H$^2$b) was estimated using the following formula and categorized as low (<0.30), medium (0.30–0.60), or high (>0.60):

$$H^2b = \sigma^2g \Big/ \left[\sigma^2g + \frac{\sigma^2ge}{e} + \frac{\sigma^2error}{re}\right], \tag{1}$$

where $\sigma^2g$ is the genotypic variance, $\sigma^2ge$ is the genotype × environment interaction variance, $\sigma^2error$ is the residual variance, $r$ is the number of replications and $e$ is the number of environments (years). The significance of environmental (years) effects was tested using Wald's statistics [31]. The mean performance of the 15 traits on regions, races, and biological type were compared using the Newman–Keuls test [32,33], using the R package "*agricolae*" [34]. Association between 15 traits was estimated by correlation coefficient and tested for significance in R software using "*corrplot*" package [35]. Gower's phenotypic distance matrix was estimated and hierarchical clustering was undertaken following Ward.D2 method [36] using the R package "*vegan*" [37] and "*cluster*" [38], and the cluster mean values were tested following the Newman–Keuls test. Promising finger millet accessions for forage quality traits were identified based on per se performance.

## 3. Results

### 3.1. ReML Variance Components Analysis

ReML variance components analysis of the agronomic and forage quality data on 314 accessions revealed highly significant genotypic variance components in individual years as well as in pooled data over years (Table 1). Similarly, the g × e variance components were significant for all the traits except CP, and their values were lower than the genotypic variance components for all the traits. Wald's statistics was used to estimate the environmental effects and found to be significant for all the traits.

**Table 1.** Estimated variance components for agronomic and forage quality traits of finger millet accessions evaluated in 2018 and 2019 at ICRISAT, Patancheru, India.

| Trait | 2018 | | 2019 | | Pooled | | | |
|---|---|---|---|---|---|---|---|---|
| | $\sigma^2g$ [#] | $\sigma^2error$ | $\sigma^2g$ | $\sigma^2error$ | $\sigma^2g$ | $\sigma^2ge$ | Wald Statistic for Years | $\sigma^2error$ |
| Days to 50% flowering | 89.01 ** | 2.076 | 54.095 ** | 2.545 | 65.996 ** | 5.52 ** | 8.85 * | 2.411 |
| Days to maturity | 106.442 ** | 4.599 | 57.588 ** | 4.366 | 73.099 ** | 9.023 ** | 16.52 ** | 4.526 |
| Grain yield (kg ha$^{-1}$) | 483,310 ** | 182,705 | 213,483 ** | 98,715 | 185,838 ** | 164,167 ** | 465.66 ** | 150,149 |
| Stover yield (kg ha$^{-1}$) | 4,536,125 ** | 2,344,396 | 1,140,943 ** | 695,538 | 1,791,173 ** | 1,015,857 ** | 839.20 ** | 1,562,927 |
| Harvest index | 0.002493 ** | 0.00159 | 0.002178 ** | 0.00112 | 0.001738 ** | 0.000553 ** | 10.46 ** | 0.001 |
| Basal tillers number | 0.9857 ** | 0.484 | 0.5001 ** | 0.424 | 0.4667 ** | 0.2728 ** | 1285.83 ** | 0.476 |
| Plant height (cm) | 125.41 ** | 43.22 | 123.52 ** | 36.04 | 105.35 ** | 17.98 ** | 153.17 ** | 43.340 |
| Dry matter (Dm) (%) | 0.0938 ** | 0.222 | 0.0404 ** | 0.254 | 0.0242 ** | 0.0445 ** | 898.19 ** | 0.517 |
| Ash content (%) | 0.3439 ** | 0.733 | 0.752 ** | 4.767 | 0.248 ** | 0.297 ** | 2276.71 ** | 2.794 |
| Crude protein (CP) (%) | 0.2856 ** | 0.511 | 0.1049 ** | 0.597 | 0.1597 ** | 0.0051 ns | 289.64 ** | 0.713 |
| Neutral detergent fiber (NDF) (%) | 2.15 ** | 3.654 | 1.616 ** | 5.376 | 0.963 ** | 0.902 ** | 895.79 ** | 4.575 |
| Acid detergent fiber (ADF) (%) | 2.472 ** | 2.945 | 1.283 ** | 1.682 | 1.195 ** | 0.413 ** | 1315.96 ** | 3.272 |
| Acid detergent lignin (ADL) (%) | 0.04788 ** | 0.0699 | 0.03828 ** | 0.0964 | 0.03183 ** | 0.00832 * | 2184.52 ** | 0.092 |
| Metabolizable energy (ME) (MJ/kg Dm) | 0.03261 ** | 0.0587 | 0.016 ** | 0.112 | 0.01766 ** | 0.00582 * | 16.48 ** | 0.090 |
| In vitro organic matter digestibility (IVOMD) (%) | 1.118 ** | 2.022 | 0.557 ** | 3.639 | 0.555 ** | 0.249 * | 73.63 ** | 2.992 |

*, ** Significant at $p \leq 0.05$, 0.01 probability levels, respectively; ns—not significant; [#] $\sigma^2g$—genotypic variance; $\sigma^2ge$—genotypic × environment variance, and $\sigma^2error$—error variance.

### 3.2. Genetic Variability and Heritability

In the individual years, all the agronomic traits differed significantly between the two years, except for days to 50% flowering and days to maturity (Table 2). The average days to 50% flowering (74 days in 2018 and 75 days in 2019) and days to maturity (108 in 2018 and 107 in 2019) were similar in both years. Grain yield and stover yield in 2019 (1865 kg ha$^{-1}$, 5598 kg ha$^{-1}$) were lower than in 2018 (2666 kg ha$^{-1}$, 8559 kg ha$^{-1}$). Similarly, the forage quality traits differed significantly between two years (Table 2). The H$^2$b of all traits varied from 0.14 to 0.98 in 2018 and 0.14 to 0.96 in 2019. All the agronomic traits showed high heritability in both the years except for basal tillers in 2019 which showed moderate heritability (0.54). Among forage quality traits, Dm (0.14) in both years and ash (0.28), NDF (0.23), ADL (0.28) in 2019 had low heritability whereas the remaining traits had moderate heritability in both the years.

**Table 2.** Mean, range, and heritability on agronomic and forage quality traits of 314 finger millet accessions evaluated in 2018 and 2019 at ICRISAT, Patancheru, India.

| Trait | Season | Mean | Range | Heritability (H²b) | # CV% | SED | LSD (p ≤ 0.05) |
|---|---|---|---|---|---|---|---|
| Days to 50% flowering | Pooled | 74 ± 0.47 | 51–97 | 0.95 | 2.08 | 1.27 | 2.49 |
| | 2018 | 74 ± 0.53 a † | 50–99 | 0.98 | 1.94 | 1.18 | 2.31 |
| | 2019 | 75 ± 0.41 a | 53–97 | 0.96 | 2.13 | 1.30 | 2.56 |
| Days to maturity | Pooled | 107 ± 0.5 | 85–130 | 0.93 | 1.98 | 1.74 | 3.41 |
| | 2018 | 108 ± 0.58 a | 84–134 | 0.96 | 1.99 | 1.75 | 3.44 |
| | 2019 | 107 ± 0.42 a | 86–128 | 0.93 | 1.96 | 1.71 | 3.35 |
| Grain yield (kg ha⁻¹) | Pooled | 2267 ± 28 | 810–3698 | 0.63 | 17.09 | 316 | 620 |
| | 2018 | 2666 ± 37 a | 615–4416 | 0.73 | 16.03 | 349 | 685 |
| | 2019 | 1865 ± 24 b | 731–3331 | 0.68 | 16.85 | 256 | 503 |
| Stover yield (kg ha⁻¹) | Pooled | 7078 ± 81 | 2890–10,779 | 0.70 | 17.66 | 1020 | 2001 |
| | 2018 | 8559 ± 111 a | 2998–13,423 | 0.66 | 17.89 | 1250 | 2454 |
| | 2019 | 5598 ± 55 b | 2952–8944 | 0.62 | 14.90 | 680 | 1336 |
| Harvest index | Pooled | 0.25 ± 0.002 | 0.12–0.37 | 0.77 | 15.68 | 0.03 | 0.06 |
| | 2018 | 0.24 ± 0.003 b | 0.12–0.39 | 0.61 | 16.50 | 0.03 | 0.06 |
| | 2019 | 0.25 ± 0.002 a | 0.11–0.36 | 0.66 | 13.36 | 0.03 | 0.05 |
| Basal tillers number | Pooled | 4 ± 0.04 | 3–7 | 0.68 | 15.69 | 0.56 | 1.10 |
| | 2018 | 5 ± 0.05 a | 4–9 | 0.67 | 13.03 | 0.57 | 1.12 |
| | 2019 | 3 ± 0.04 b | 2–6 | 0.54 | 18.87 | 0.53 | 1.04 |
| Plant height (cm) | Pooled | 120 ± 0.59 | 84–143 | 0.87 | 5.50 | 5.38 | 10.54 |
| | 2018 | 123 ± 0.6 a | 85–150 | 0.74 | 5.36 | 5.37 | 10.54 |
| | 2019 | 117 ± 0.6 b | 78–138 | 0.77 | 5.13 | 4.90 | 9.62 |
| Dry matter (Dm) (%) | Pooled | 90.42 ± 0 | 90.39–90.44 | 0.18 | 0.80 | 0.59 | 1.15 |
| | 2018 | 90.84 ± 0.01 a | 90.15–91.78 | 0.14 | 0.52 | 0.38 | 0.76 |
| | 2019 | 89.99 ± 0.01 b | 89.64–90.36 | 0.14 | 0.56 | 0.41 | 0.81 |
| Ash content (%) | Pooled | 12.42 ± 0.02 | 11.47–14.17 | 0.29 | 13.46 | 1.36 | 2.68 |
| | 2018 | 10.31 ± 0.03 b | 9.12–11.69 | 0.32 | 8.30 | 0.70 | 1.37 |
| | 2019 | 14.53 ± 0.03 a | 13.44–16.75 | 0.28 | 15.02 | 1.78 | 3.50 |
| Crude protein (CP) (%) | Pooled | 7.23 ± 0.02 | 6.47–8.15 | 0.57 | 11.68 | 0.69 | 1.35 |
| | 2018 | 7.56 ± 0.02 a | 6.50–8.82 | 0.36 | 9.45 | 0.58 | 1.15 |
| | 2019 | 6.9 ± 0.01 b | 6.42–7.60 | 0.30 | 11.19 | 0.63 | 1.24 |
| Neutral detergent fiber (NDF) (%) | Pooled | 64.94 ± 0.05 | 62.36–67.73 | 0.44 | 3.29 | 1.75 | 3.42 |
| | 2018 | 66.8 ± 0.07 a | 62.96–69.55 | 0.37 | 2.86 | 1.56 | 3.06 |
| | 2019 | 63.08 ± 0.05 b | 60.03–65.39 | 0.23 | 3.68 | 1.89 | 3.72 |
| Acid detergent fiber (ADF) (%) | Pooled | 35.87 ± 0.06 | 33.07–38.37 | 0.61 | 5.04 | 1.48 | 2.90 |
| | 2018 | 37.64 ± 0.08 a | 34.33–40.58 | 0.46 | 4.56 | 1.40 | 2.75 |
| | 2019 | 34.1 ± 0.05 b | 31.43–36.80 | 0.43 | 3.80 | 1.06 | 2.08 |

**Table 2.** *Cont.*

| Trait | Season | Mean | Range | Heritability (H²b) | # CV% | SED | LSD (*p* ≤ 0.05) |
|---|---|---|---|---|---|---|---|
| Acid detergent lignin (ADL) (%) | Pooled | 4.39 ± 0.01 | 3.95–4.80 | 0.62 | 6.90 | 0.25 | 0.48 |
| | 2018 | 4.76 ± 0.01 a | 4.31–5.19 | 0.41 | 5.56 | 0.22 | 0.42 |
| | 2019 | 4.02 ± 0.01 b | 3.65–4.36 | 0.28 | 7.71 | 0.25 | 0.50 |
| Metabolizable energy (ME) (MJ/kg Dm) | Pooled | 6.57 ± 0.01 | 6.18–6.89 | 0.50 | 4.56 | 0.24 | 0.48 |
| | 2018 | 6.54 ± 0.01 b | 6.08–7.09 | 0.36 | 3.70 | 0.20 | 0.39 |
| | 2019 | 6.6 ± 0 a | 6.27–6.77 | 0.33 | 5.07 | 0.27 | 0.54 |
| In vitro organic matter digestibility (IVOMD) (%) | Pooled | 47.28 ± 0.04 | 45.21–49.09 | 0.47 | 3.66 | 1.41 | 2.77 |
| | 2018 | 46.9 ± 0.05 b | 43.87–49.84 | 0.36 | 3.03 | 1.16 | 2.28 |
| | 2019 | 47.66 ± 0.02 a | 45.83–48.75 | 0.33 | 4.00 | 1.56 | 3.06 |

[†] Mean followed by the same letters are not significant at *p* ≤ 0.05, and means followed by different letters are significant at *p* ≤ 0.05. [#] CV—coefficient of variation; SED—standard error of a difference; LSD—least significant difference.

Based on the pooled data over two years, the finger millet accessions matured in 85 to 130 days and produced a grain yield of 810 to 3698 kg ha$^{-1}$. Plant height and basal tillers number are the important forage traits that varied from 84 to 143 cm and 3 to 7, respectively. Stover yield (dry) of the tested accessions varied from 2890 to 10,779 kg ha$^{-1}$ with a mean of 7078 kg ha$^{-1}$. The forage quality traits namely Dm (90.39–90.44%), ME (6.18–6.89 MJ/kg Dm), and ADL (3.95–4.80%) had a narrow range of variation, while the remaining traits namely ash content (11.47–14.17%), CP (6.47–8.15%), NDF (62.36–67.73%), ADF (33.07–38.37%), and IVOMD (45.21–49.09%) showed moderate variability (Table 2). The estimated heritability of all the traits varied from low (0.18) to high (0.95) (Table 2). All the agronomic traits showed high heritability. Among the forage quality traits, ADF (0.61) and ADL (0.62) showed high H$^2$b while CP (0.57), NDF (0.44), ME (0.50), IVOMD (0.47) showed moderate H$^2$b, and Dm (0.18) and Ash (0.29) showed low H$^2$b.

### 3.3. Mean Comparison among Races, Regions, and Biological Status

The means of both agronomic and forage quality traits were compared among different races, regions of origin, and biological status (Table 3). Among the races, the race *vulgaris* flowered significantly earlier (73 days) than the other races and matured in 105 days, while race *plana* and *elongata* matured up to 112 days. Race *plana* had a higher stover yield (7721 kg ha$^{-1}$), ME (6.63 MJ/kg Dm), and IVOMD (47.58%) than race *vulgaris* (stover yield-6833 kg ha$^{-1}$, ME-6.55 MJ/kg Dm, IVOMD-47.19%), low in NDF (64.78%) than in race *elongata* (65.27%). Race *compacta* had low ADL (4.33%), which differed significantly from the race *elongata* (4.42%). Among regions, we compared means of agronomic and forage quality traits only between Africa and Asia while accessions from Europe, North America, and unknown origin were not included because only a few accessions (<10) were available. Accessions from Africa were late flowering (seven days later than Asian accessions), produced tall plants (123 cm) with a low number of tillers (4 tillers), and higher stover yield (7440 kg ha$^{-1}$) than accessions from Asia (Table 3). For forage traits, accessions from Asia were high in crude protein (7.36%), while low in NDF (64.78%) (low value is desirable), and energy sources such as ME (6.54 MJ/kg Dm) and IVOMD (47.12%) than African accessions. There was no significant difference between accessions of Africa and Asia for average grain yield, Dm, ADF, and ADL. Similarly, landraces and breeding lines differed significantly for 10 of 15 traits, except for stover yield, Dm, ADL, ME, and IVOMD. On average, breeding lines matured early and produced higher grain yield (2583 kg ha$^{-1}$) with higher HI (0.27), ash content (12.55%), and CP (7.31%), and lower ADF (35.58%) and NDF (64.67%) than landraces.

### 3.4. Correlation among Forage Quality Traits and Agronomic Traits

Correlation coefficients were estimated using pooled data over two years for the agronomic and forage quality traits (Table 4). Among the forage quality traits, stronger positive correlations were observed between fiber fractions (ADF, NDF, ADL); ash and CP; IVOMD and ME. Forage quality positive traits such as IVOMD and ME had a significantly negative correlation with all other forage quality traits. The CP content was significant and negatively associated with both NDF and ADL. Dm had a significantly positive association with ash content, ADF, ADL, and a non-significant association with CP and NDF. Ash content showed a significantly negative correlation with NDF, while it was non-significant association with ADF and ADL.

**Table 3.** Mean comparison on races, region, and biological status of 314 finger millet accessions on agronomic and forage quality traits.

| Trait # | Race | | | | Region | | Biological Status | |
|---|---|---|---|---|---|---|---|---|
| | *Compacta* | *Elongata* | *Plana* | *Vulgaris* | **Africa** | **Asia** | **Breeding Lines** | **Landraces** |
| DF | 75 ± 1.82 ab † | 78 ± 1.43 a | 79 ± 0.95 a | 73 ± 0.56 b | 78 ± 0.52 a | 71 ± 0.72 b | 70 ± 1.14 b | 75 ± 0.5 a |
| DM | 108 ± 1.94 ab | 111 ± 1.55 a | 112 ± 1.04 a | 105 ± 0.59 b | 111 ± 0.56 a | 103 ± 0.75 b | 103 ± 1.19 b | 108 ± 0.53 a |
| GY | 2261 ± 86 a | 2149 ± 120 a | 2295 ± 63 a | 2267 ± 331 a | 2252 ± 34 a | 2307 ± 48 a | 2583 ± 57 a | 2206 ± 29 b |
| SY | 7374 ± 300 ab | 7227 ± 241 ab | 7721 ± 169 a | 6833 ± 100 b | 7440 ± 83 a | 6620 ± 142 b | 7184 ± 253 a | 7058 ± 83 a |
| HI | 0.24 ± 0.01 a | 0.23 ± 0.01 a | 0.23 ± 0.01 a | 0.25 ± 0.02 a | 0.23 ± 0.01 b | 0.26 ± 0.01 a | 0.27 ± 0.01 a | 0.24 ± 0.01 b |
| BTN | 4 ± 0.10 a | 4 ± 0.14 a | 4 ± 0.06 a | 5 ± 0.05 b | 4 ± 0.05 b | 5 ± 0.06 a | 5 ± 0.09 a | 4 ± 0.04 b |
| PH | 118 ± 1.96 a | 121 ± 2.15 a | 124 ± 1.45 a | 119 ± 0.71 a | 123 ± 0.65 a | 115 ± 0.88 b | 116 ± 1.48 b | 121 ± 0.63 a |
| Dm | 90.42 ± 0.001 a | 90.42 ± 0.001 a | 90.42 ± 0.001 a | 90.42 ± 0.001 a | 90.42 ± 0.001 a | 90.42 ± 0.001 a | 90.42 ± 0.001 a | 90.42 ± 0.001 a |
| Ash | 12.46 ± 0.08 a | 12.26 ± 0.08 a | 12.41 ± 0.06 a | 12.45 ± 0.03 a | 12.28 ± 0.03 b | 12.60 ± 0.04 a | 12.55 ± 0.05 a | 12.40 ± 0.03 b |
| CP | 7.25 ± 0.05 a | 7.16 ± 0.06 a | 7.12 ± 0.04 a | 7.27 ± 0.02 a | 7.13 ± 0.02 b | 7.36 ± 0.02 a | 7.31 ± 0.03 a | 7.22 ± 0.02 b |
| NDF | 64.58 ± 0.19 b | 65.27 ± 0.21 a | 64.78 ± 0.14 b | 65.00 ± 0.06 ab | 65.07 ± 0.08 a | 64.78 ± 0.08 b | 64.67 ± 0.13 b | 64.99 ± 0.06 a |
| ADF | 35.55 ± 0.21 a | 36.08 ± 0.21 a | 35.64 ± 0.14 a | 35.96 ± 0.07 a | 35.89 ± 0.08 a | 35.86 ± 0.09 a | 35.58 ± 0.15 b | 35.93 ± 0.06 a |
| ADL | 4.33 ± 0.03 b | 4.42 ± 0.03 a | 4.35 ± 0.02 ab | 4.41 ± 0.01 ab | 4.40 ± 0.01 a | 4.39 ± 0.01 a | 4.37 ± 0.02 a | 4.40 ± 0.01 a |
| ME | 6.59 ± 0.02 ab | 6.58 ± 0.02 ab | 6.63 ± 0.02 a | 6.55 ± 0.01 b | 6.60 ± 0.01 a | 6.54 ± 0.01 b | 6.54 ± 0.01 a | 6.58 ± 0.01 a |
| IVOMD | 47.42 ± 0.11 ab | 47.28 ± 0.10 ab | 47.58 ± 0.09 a | 47.19 ± 0.04 b | 47.41 ± 0.05 a | 47.12 ± 0.05 b | 47.13 ± 0.08 a | 47.31 ± 0.04 a |

# DF = Days to 50% flowering, DM = Days to maturity, GY = Grain yield (kg ha$^{-1}$), SY = Stover yield (kg ha$^{-1}$), HI = Harvest index, BTN = Basal tillers number, PH = Plant height (cm), Dm = Dry matter (%), Ash content (%), CP = Crude protein (%), NDF = Neutral detergent fiber (%), ADF = Acid detergent fiber (%), ADL = Acid detergent lignin (%), ME = Metabolizable energy (MJ/kg Dm), IVOMD = In vitro organic matter digestibility (%); † Mean followed by the same letters are not significant at $p \leq 0.05$, and means followed by different letters are significant at $p \leq 0.05$.

**Table 4.** Correlation among forage quality and agronomic traits in 314 finger millet accessions.

| Trait [#] | DM | GY | SY | HI | BTN | PH | Dm | Ash | CP | NDF | ADF | ADL | ME | IVOMD |
|---|---|---|---|---|---|---|---|---|---|---|---|---|---|---|
| DF | 0.996 ** | 0.024 | 0.647 ** | −0.621 ** | −0.408 ** | 0.405 ** | −0.379 ** | −0.395 ** | −0.325 ** | 0.034 | −0.211 ** | −0.251 ** | 0.405 ** | 0.358 ** |
| DM | | 0.021 | 0.643 ** | −0.617 ** | −0.410 ** | 0.392 ** | −0.383 ** | −0.398 ** | −0.319 ** | 0.034 | −0.209 ** | −0.248 ** | 0.403 ** | 0.356 ** |
| GY | | | 0.453 ** | 0.531 ** | 0.140 * | 0.019 | −0.071 | −0.044 | −0.232 ** | −0.043 | −0.137 * | −0.046 | 0.056 | 0.005 |
| SY | | | | −0.469 ** | −0.231 ** | 0.540 ** | −0.216 ** | −0.396 ** | −0.500 ** | −0.185 ** | −0.442 ** | −0.378 ** | 0.479 ** | 0.384 ** |
| HI | | | | | 0.400 ** | −0.516 ** | 0.143 * | 0.343 ** | 0.225 ** | 0.100 | 0.249 ** | 0.308 ** | −0.379 ** | −0.344 ** |
| BTN | | | | | | −0.290 ** | 0.039 | 0.123 * | 0.070 | 0.175 ** | 0.301 ** | 0.283 ** | −0.357 ** | −0.364 ** |
| PH | | | | | | | 0.000 | −0.423 ** | −0.521 ** | 0.163 ** | 0.000 | 0.011 | 0.283 ** | 0.183 ** |
| Dm | | | | | | | | 0.195 ** | 0.082 | −0.043 | 0.113 * | 0.139 * | −0.150 ** | −0.151 ** |
| Ash | | | | | | | | | 0.658 ** | −0.374 ** | −0.054 | −0.095 | −0.482 ** | −0.338 ** |
| CP | | | | | | | | | | −0.287 ** | −0.108 | −0.119 * | −0.480 ** | −0.313 ** |
| NDF | | | | | | | | | | | 0.777 ** | 0.843 ** | −0.395 ** | −0.493 ** |
| ADF | | | | | | | | | | | | 0.845 ** | −0.539 ** | −0.566 ** |
| ADL | | | | | | | | | | | | | −0.646 ** | −0.730 ** |
| ME | | | | | | | | | | | | | | 0.976 ** |

[#] DF = Days to 50% flowering, DM = Days to maturity, GY = Grain yield (kg ha$^{-1}$), SY = Stover yield (kg ha$^{-1}$), HI = Harvest index, BTN = Basal tillers number, PH = Plant height (cm), Dm = Dry matter (%), Ash = Ash content (%), CP = Crude protein (%), NDF = Neutral detergent fiber (%), ADF = Acid detergent fiber (%), ADL = Acid detergent lignin (%), ME = Metabolizable energy (MJ/kg Dm), IVOMD = In vitro organic matter digestibility (%); *, ** Significant at $p \leq 0.05$, 0.01 probability levels, respectively.

Correlation between forage quality traits with agronomic traits showed that days to 50% flowering and maturity tended to be significantly positively associated with ME, and IVOMD while this relationship was inversely related (significant) with Dm, ash, CP, ADF, and ADL and non-significant association with NDF. Stover yield had a significantly positive correlation with ME and IVOMD while this relationship was negative but significant with remaining forage quality traits. HI had a positive significant correlation with Dm, ash, CP, ADF, and ADL, whereas significantly negatively correlated with ME and IVOMD. Similarly, basal tillers number showed a significant negative association with IVOMD and ME, and a significantly positive association with fiber fractions and ash content. A significant positive association was observed between plant height with NDF, ME, and IVOMD, and a negative significant correlation with ash and CP.

Since the performance of finger millet germplasm differed based on regions and biological status, correlation coefficients were performed separately on landraces, breeding lines, African, and Asian accessions and their results were compared to gain knowledge on any effect of regions and biological status on forage quality traits. Correlations of agronomic and forage quality traits in the breeding lines, landraces, African, and Asian germplasms differed considerably (Table S2a,b). For example, correlations between agronomic and forage quality traits in the landraces indicated that the traits namely Dm, Ash, and crude protein contents were significantly negatively associated with days to 50% flowering, days to maturity, and stover yield, while these correlations were non-significant in the breeding lines. The Ash content in the landraces was significantly negatively associated with all the agronomic traits except grain yield. Similarly, in Asia, correlations between days to 50% flowering, maturity and grain yield with ADF and ADL were significantly negative while these correlations were non-significant in the African germplasm.

Among the forage quality traits, Dm showed a significant positive correlation with ash content in the landraces while non-significant in the breeding lines (Table S2a). Similarly, CP, NDF, ADF, and ADL showed a significant positive correlation with Dm while a significant negative correlation with ME in the breeding lines; however, all these traits showed non-significant association in the landraces. The CP in the landraces showed a significantly negative correlation with NDF, ME, and IVOMD while these were non-significant in the breeding lines. The ash content in the landraces was significantly and negatively associated with NDF and IVOMD while it was non-significant in the breeding lines. The remaining forage quality traits in both landraces and breeding lines had similar associations. The correlations between Dm with Ash, ADF, and ADL were significantly positive, while ME and IVOMD were negatively correlated with Dm in the accessions from Asia whereas it was non-significant in accessions from Africa (Table S2b). The NDF with Dm and CP with NDF, ADF, and ADL were non-significant in Asia accessions while these relationships were significant and negative in Africa accessions. All the remaining correlations among the forage quality traits were similar in Asian and African accessions.

*3.5. Cluster Analysis*

The cluster analysis of finger millet accessions using both agronomic and forage quality traits grouped the accessions into 5 distinct clusters (Figure 2). The number of accessions in each cluster varied from 41 in cluster 3 to 72 in cluster 4 (Table 5). Cluster 1 consisted of 64 accessions, while cluster 2 and cluster 5 had 66 and 71 accessions, respectively. Among 5 clusters, cluster 3 mainly consists of accessions from Asia, particularly from India and Nepal, the remaining clusters had a mixture of accessions from Africa and Asia. The trait mean values of the 5 clusters significantly differed among each other. On average, accessions in clusters 3 and 5 were early flowering accessions, while those in clusters 1, 2, and 4 were late-flowering accessions (Table 5). Interestingly, accessions in cluster 1 were found to have desirable forage yield and quality traits (higher grain and stover yield with a low value of fiber components and higher stover digestibility). Most of the top performing accessions for forage traits were in cluster 1 (Table 6). On average, the phenotypic within-cluster distance was higher in cluster 1 (0.163), followed by clusters 3

and 5 which had a similar value (0.153), while cluster 4 had the lowest mean distance of 0.137. The highest average distance between clusters was observed between clusters 1 and 3 (0.303) while the lowest distance was observed between cluster 2 and cluster 4 (0.173) (Table 6).

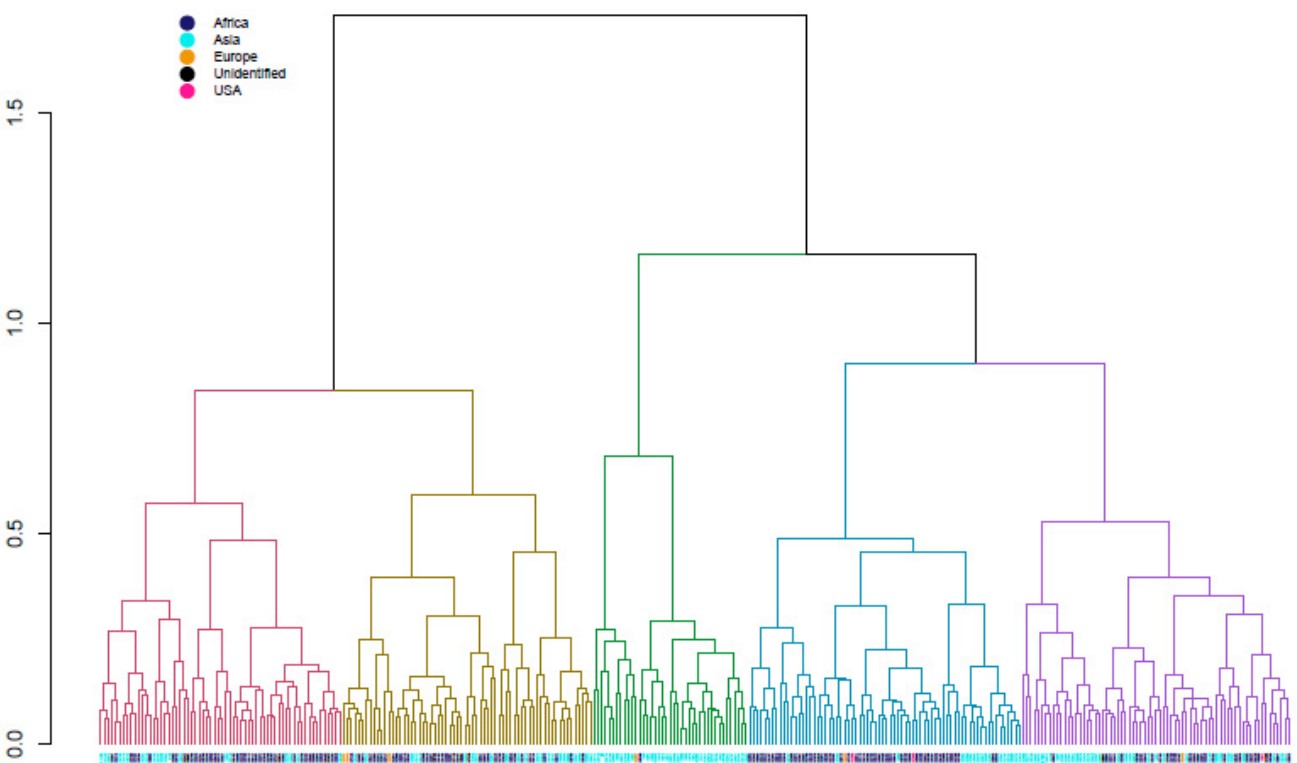

**Figure 2.** Dendrogram showing the clustering of 314 finger millet accessions based on forage quality and agronomic traits. Coloured branches indicate clusters 1 to 5 (from left to right).

**Table 5.** Mean comparison among five clusters for agronomic and forage quality traits in 314 finger millet accessions.

| Trait [#] | Cluster 1 (64) | | Cluster 2 (66) | | Cluster 3 (41) | | Cluster 4 (72) | | Cluster 5 (71) | |
|---|---|---|---|---|---|---|---|---|---|---|
| | Mean | Range | Mean | Range | Mean | Range | Mean | Range | Mean | Range |
| DF | 77 b [†] | 67–91 | 83 a | 65–97 | 62 d | 51–75 | 76 b | 67–88 | 70 c | 60–76 |
| DM | 110 b | 99–125 | 116 a | 96–130 | 94 d | 85–107 | 109 b | 99–122 | 103 c | 92–109 |
| GY | 2368 a | 1554–3214 | 2006 b | 810–2909 | 2059 b | 1233–2924 | 2377 a | 1235–3538 | 2427 a | 1324–3698 |
| SY | 8290 a | 5957–10,779 | 7557 b | 3760–10,123 | 5012 d | 2891–6617 | 7244 b | 5805–10,186 | 6567 c | 4726–9478 |
| HI | 0.22 d | 0.17–0.27 | 0.21 e | 0.12–0.29 | 0.29 a | 0.21–0.37 | 0.25 c | 0.16–0.32 | 0.27 b | 0.2–0.34 |
| BTN | 4 d | 3–6 | 4 cd | 3–7 | 5 a | 4–7 | 5 b | 3–7 | 4 bc | 3–6 |
| PH | 124 a | 102–138 | 121 ab | 95–143 | 109 c | 84–122 | 124 a | 110–140 | 117 b | 93–132 |
| Dm | 90.42 b | 90.40–90.44 | 90.41 c | 90.40–90.43 | 90.42 a | 90.41–90.44 | 90.42 b | 90.40–90.43 | 90.42 b | 90.41–90.44 |
| Ash | 12.44 b | 11.64–13.78 | 12.18 c | 11.49–12.85 | 12.74 a | 11.82–14.17 | 12.28 c | 11.52–13.18 | 12.60 a | 11.47–13.57 |
| CP | 7.18 c | 6.58–8.05 | 7.17 c | 6.47–7.88 | 7.50 a | 7.09–8.15 | 7.07 c | 6.57–7.53 | 7.34 b | 6.77–7.91 |
| NDF | 63.88 d | 62.37–65.41 | 65.04 b | 63.8–66.42 | 65.31 b | 62.94–67.09 | 65.91 a | 64.07–67.74 | 64.61 c | 63.25–65.74 |
| ADF | 34.67 c | 33.08–36.16 | 35.73 b | 34.51–37.17 | 36.65 a | 34.56–37.54 | 36.72 a | 35.28–38.37 | 35.78 b | 34.26–37.04 |
| ADL | 4.20 c | 3.96–4.49 | 4.35 b | 4.18–4.59 | 4.54 a | 4.16–4.79 | 4.54 a | 4.4–4.81 | 4.37 b | 4.16–4.62 |
| ME | 6.68 a | 6.46–6.9 | 6.62 b | 6.47–6.76 | 6.44 d | 6.19–6.57 | 6.53 c | 6.37–6.69 | 6.55 c | 6.38–6.79 |
| IVOMD | 47.90 a | 46.77–49.09 | 47.56 b | 46.69–48.11 | 46.56 e | 45.21–47.53 | 46.94 d | 45.82–47.93 | 47.23 c | 46.21–48.57 |

[#] DF = Days to 50% flowering, DM = Days to maturity, GY = Grain yield (kg ha$^{-1}$), SY = Stover yield (kg ha$^{-1}$), HI = Harvest index, BTN = Basal tillers number, PH = Plant height (cm), Dm = Dry matter (%), Ash content (%), CP = Crude protein (%), NDF = Neutral detergent fiber (%), ADF = Acid detergent fiber (%), ADL = Acid detergent lignin (%), ME = Metabolizable energy (MJ/kg Dm), IVOMD = In vitro organic matter digestibility (%). [†] Mean followed by the same letters are not significant at $p \leq 0.05$, and means followed by different letters are significant at $p \leq 0.05$. Values in the parenthesis represent the number of accessions present in each cluster.

**Table 6.** Gower's phenotypic distance (mean) within and among five clusters.

| Cluster | 1 | 2 | 3 | 4 | 5 |
|---|---|---|---|---|---|
| 1 | **0.163** [†] | 0.203 | 0.303 | 0.219 | 0.284 |
| 2 | | **0.144** | 0.193 | 0.173 | 0.191 |
| 3 | | | **0.153** | 0.217 | 0.184 |
| 4 | | | | **0.137** | 0.185 |
| 5 | | | | | **0.153** |

[†] Values in bold indicate the within-cluster mean of five clusters.

### 3.6. Promising Trait-Specific Accessions

Promising trait-specific accessions for forage quality traits were identified and compared with the mean value of important agronomic traits such as days to 50% flowering and grain yield. All the identified accessions were significantly higher than the overall mean for the respective traits. The top 10 accessions with desirable values for each forage quality trait and multiple quality traits were identified (Tables 7 and 8). Among the multiple quality traits, IE 5435 was found to be a promising source for 7 forage quality traits while IE 3723 and IE 3821 for 6 traits. The remaining accessions were superior for 3 or 4 forage quality traits. The top 10 promising accessions with multiple forage quality traits had a wide range of days to 50% flowering from 61 to 91 days and yield ranging from 1596 to 2787 kg ha$^{-1}$.

**Table 7.** Top 10 finger millet accessions with the desirable value of each forage quality trait.

| Trait | Class | Top Ten Accessions with High Trait Values | Days to 50% Flowering (Days) | Grain Yield (kg ha$^{-1}$) |
|---|---|---|---|---|
| Stover yield (SY) (kg ha$^{-1}$) | High | IE 2039, IE 2045, IE 2939, IE 3473, IE 4699, IE 4700, IE 4701, IE 4707, IE 5435, MR 6 | 76–88 | 2088–3530 |
| Dm Dry matter (Dm) (%) | High | IE 501, IE 588, IE 615, IE 872, IE 1055, IE 2217, IE 3769, IE 5165, IE 5407, IE 5435 | 51–79 | 1616–3698 |
| Ash (%) | Low * | IE 2066, IE 2586, IE 2789, IE 2911, IE 4646, IE 3025, IE 3443, IE 4497, IE 6326, IE 6537 | 60–95 | 1233–3342 |
| Crude protein (CP) (%) | High | IE 588, IE 2030, IE 2437, IE 2957, IE 4660, IE 4700, IE 4816, IE 5736, IE 5817, IE 5956 | 51–85 | 1029–3140 |
| Neutral detergent fiber (NDF) (%) | Low * | IE 2437, IE 3129, IE 3130, IE 3723, IE 3821, IE 3947, IE 4654, IE 4737, IE 5435, IE 8602 | 55–81 | 1596–3313 |
| Acid detergent fiber (ADF) (%) | Low * | IE 2045, IE 3128, IE 3129, IE 3723, IE 3821, IE 3947, IE 4654, IE 4699, IE 5249, IE 5435 | 66–84 | 2088–3121 |
| Acid detergent lignin (ADL) (%) | Low * | IE 953, IE 2437, IE 2760, IE 3128, IE 3129, IE 3130, IE 3723, IE 3821, IE 4654, IE 5364 | 66–90 | 1596–3002 |
| Metabolizable energy (ME) (MJ/kg Dm) | High | IE 2066, IE 2341, IE 2760, IE 3614, IE 3723, IE 3821, IE 3952, IE 4646, IE 5435, IE 8790 | 60–90 | 1909–2870 |
| In vitro organic matter digestibility (IVOMD) (%) | High | IE 2217, IE2341, IE 2760, IE 3129, IE 3614, IE 3723, IE 3821, IE 4646, IE 5435, IE 8790 | 60–90 | 1909–2870 |

* Low quantity is desirable.

**Table 8.** Finger millet accessions with desirable multiple forage quality traits.

| Accession | Cluster No. | SY kg ha$^{-1}$ | Dm (%) | Ash (%) | CP (%) | NDF (%) | ADF (%) | ADL (%) | ME (MJ/ kg Dm) | IVOMD (%) | DF [#] (days) | GY kg ha$^{-1}$ | Region | Biological Type |
|---|---|---|---|---|---|---|---|---|---|---|---|---|---|---|
| IE 2066 | 5 | 7368 | 90.41 | **11.47** | 6.76 | **63.28** | 35.07 | 4.19 | **6.78** | **48.39** | 61 | 2496 | Asia | Improved cultivar |
| IE 2437 | 1 | 6623 | 90.42 | 13.06 | **8.05** | **62.97** | 34.77 | **4.01** | 6.60 | 47.90 | 81 | 1596 | Africa | Landrace |
| IE 3129 | 1 | 8924 | 90.41 | 12.14 | 7.46 | **63.09** | 33.09 | **4.03** | **6.77** | **48.51** | 80 | 2223 | Asia | Improved cultivar |
| IE 3614 | 1 | 8808 | 90.42 | 12.03 | 6.84 | 63.61 | 34.61 | **4.11** | **6.89** | **49.03** | 72 | 2118 | Unidentified | Landrace |
| IE 3723 | 1 | 9658 * | 90.43 | 12.29 | 7.18 | **62.58** | **33.61** | **4.04** | **6.89** | **49.09** | 79 | 2278 | Africa | Landrace |
| IE 3821 | 1 | **9791** | 90.42 | 12.29 | 7.13 | **62.36** | **33.07** | **3.98** | **6.85** | **48.94** | 79 | 2138 | Africa | Landrace |
| IE 3952 | 1 | **9820** | 90.42 | 12.02 | 6.58 | 64.01 | 34.33 | 4.16 | **6.79** | **48.36** | 77 | 2647 | Africa | Landrace |
| IE 4700 | 1 | **10,158** | 90.42 | 13.38 | **7.86** | **63.38** | 34.14 | 4.17 | 6.53 | 47.30 | 86 | 2787 | Asia | Improved cultivar |
| IE 5435 | 1 | **10,640** | **90.43** | 12.03 | 6.87 | **62.96** | **33.57** | **4.12** | **6.88** | **48.97** | 80 | 2088 | Africa | Landrace |
| IE 8790 | 1 | 8139 | 90.41 | 12.20 | 7.11 | 63.29 | 34.18 | **4.10** | **6.79** | **48.52** | 91 | 2022 | Africa | Landrace |

[#] DF = Days to 50% flowering, GY = Grain yield (kg ha$^{-1}$), SY = Stover yield (kg ha$^{-1}$), Dm = Dry matter (%), Ash content (%), CP = Crude protein (%), NDF = Neutral detergent fiber (%), ADF = Acid detergent fiber (%), ADL = Acid detergent lignin (%); ME = Metabolizable energy (MJ/kg Dm), IVOMD = In vitro organic matter digestibility (%). * Values in bold indicate higher levels of multiple nutrients except for Ash, NDF, ADF, and ADL were the lowest in desirable.

## 4. Discussion

### 4.1. Variance Component and Heritability

Finger millet is a climate-resilient nutrient dense crop and can contribute potentially to food, feed, and nutritional security in the changing climate scenario. Thus, research on the forage quality of finger millet stover can support promoting it as a dual-purpose crop. A wide variability present in germplasm collections provides the new diversity required for trait improvement. The ICRISAT genebank conserves a global collection of finger millet germplasm. This study utilized a diversity panel of 314 accessions originating from 23 countries to assess variability and to identify promising germplasm for use in finger millet improvement.

The ReML analysis of variance indicated significant variability present in the germplasm for forage quality traits and related agronomic traits (Table 1). The significant genotype × environment interactions were also observed for all the traits except for crude protein, indicating the significant role of the environment and their interaction with genotype on the forage quality traits. However, all the agronomic traits namely days to flowering, maturity, plant height, basal tillers number, grain, and stover yield showed higher heritability while the forage quality traits showed low to moderate except ADF (0.61) and ADL (0.62) which fall under the high heritability category. The traits with high heritability are controlled by additive genes, have minimum environmental influences, therefore, the possibility of a good response to selection [39]. However, the traits with low to moderate heritability are suggested to be complex traits and highly influenced by the environment. In our study, 9 out of 15 traits had $H^2b > 0.60$, indicating the phenotypic selection could be effectively used [23] to improve the forage quality traits in finger millet.

### 4.2. Variability and Mean Performance of Finger Millet for Forage Quality Traits

The mean performance of accessions showed a significant difference across the two years for all agronomic and forage quality traits (Table 2). The seasonal changes during the years 2018 and 2019 (Figure 1) could influence the growth and development of the finger millet crop. In particular, distribution and amount of rainfall was higher in 2019 compared to 2018, and the distribution was maximum during the flowering stage followed by in the vegetative stage. The mean and range of minimum and maximum temperature were similar in both years. Significant variability was found in the studied germplasm for forage quality traits. Baath et al. [11] reported a wider range of variation among 11 germplasms conserved at the USDA-ARS Plant Genetic Resources Conservation Unit, Griffin, GA, USA, compared to the results in this study. However, the present study was the first report in utilizing a large number of a diverse global collection of finger millet germplasm and this can provide a greater range and potential of finger millet crop residues for forage quality traits. In other crops, both a low variability for forage quality traits in wheat [21,22] and higher variability in maize [25], sorghum [40], pearl millet [26], and rice [16,17,20] were reported.

Stover yield is one of the key objectives of forage crop improvement. The total stover yield in this study ranged from 2.8 to 10.7 tones ha$^{-1}$ with a mean of 7.0 tones ha$^{-1}$. In previous studies, up to 12 tones ha$^{-1}$ under irrigated conditions [11] and up to 3 tones ha$^{-1}$ [12] under rainfed condition was reported. The mean finger millet stover yield (7.0 tones ha$^{-1}$) in this study was lower than that reported in sorghum of 11.04 tones ha$^{-1}$ [40] but comparable with rice (7.1 tones ha$^{-1}$) [20] and higher than in wheat (6.1 tones ha$^{-1}$) [41]. A good quality feed (grass) should contain >9% CP, >85% dry matter, <50% NDF, <35% ADF [42]. Forage with high CP, IVOMD, ME, and lower ADF, NDF, ADL are more desirable for the digestibility of the forage, thereby increasing animal feed intake [19,42].

Crude protein plays an important role in animal digestion because, for timely and efficient digestion, rumen microbes need at least 7% of CP in the diet. However, the contribution of crude protein from cereal crop residues in the animal diet is low [40,43,44]. In our study, the crude protein content ranged from 6.47% to 8.15% with a mean of 7.23%, which was comparable to the finger millet germplasm mean value (7.45%) [13] and it is

sufficient for better feed digestion. The range of CP in this study was low compared to another study with 11 accessions of finger millet (10.5% to 15.6%) conserved at USDA-ARS [11], but higher than in rice (4.06% to 7.80%) and wheat (3.68% to 4.43%) straw [20,22].

The mean value of the NDF in the finger millet crop residue was higher (65%) than the desirable level (<50%) [42]. However, the observed mean value of ADF was on par with the recommended level (35%) and constitute about half the portion of NDF. The mean ADF and NDF values of finger millet in this study were lower than in rice (ADF 51–52%; NDF 66.0%) [18,20] and wheat (ADF-50–51%; NDF-77.5%) [18,22]. The observed mean ADL was higher than in rice (3.9%) [20] but lower than in wheat (5.9%) [22]. The IVOMD followed by CP is the key determinant of the nutritional value of the forage [45,46]. The proportion of the feed that the animal can digest is denoted by IVOMD and summarizes all the effects and interactions of Nitrogen, ADF, NDF, and ADL [44,47]. The average IVOMD value in finger millet in this study (47%) was higher than the reported value in rice (42%) [17,20] whereas lower than in wheat (up to 48%) [21,22]. In livestock production, energy is the second major limiting factor in animal performance, next to CP and it is measured by evaluating the ME content of the feed [48,49]. The mean ME in finger millet was lower than in wheat (7.0 MJ/kg Dm) [22], and groundnut (7.9 MJ/kg Dm) [27]. Dry matter estimates can be used to compare the different feeds on the equivalent premise, monitoring the animal dry matter intake, and evaluate the moisture content of the particular feed [42]. However, the variability in our finger millet dry matter content was too narrow and the value was above >90%.

The per se performance of accessions among regions, races, and biological status showed a significant effect on forage traits (Table 3). Among regions, accessions from Asia had superior forage quality such as high protein content, and low in NDF with high grain yield than accessions from Africa. Nonetheless, stover yield was high in the African germplasm. Similarly, breeding lines had better forage quality traits than landraces and both did not differ for stover yield and energy sources.

### 4.3. Relationship among Agronomic and Forage Quality Traits

In the entire set (Table 4), the correlation between the forage quality positive traits such as IVOMD and ME was positive and significant with stover yield, and both were non-significant and negligible with grain yield indicating that it is possible to improve stover yield and grain yield in the background of higher energy sources and digestibility. The CP is one of the key factors in forage quality traits to determine the feed value but the correlations between CP with IVOMD and ME, and also with grain yield and stover yield were negatively significant. It is to be notified that cereals' crop residues were mainly used as an energy source for livestock rather than a protein supplement. Thus, reduction in the CP concentration during the forage quality traits improvement in finger millet can be compensated with the N richer supplementation. CP with IVOMD was positive and significant in finger millet [13] and Napier grass [50], whereas it was negative and non-significant in sorghum [28]. The negatively significant correlation between CP and grain yield was also reported in rice and wheat [20,51]. In this study, significant negative associations were observed between in vitro digestibility traits (IVOMD and ME) and stover yield with fiber components showing increases in cellular components reducing the stover biomass and in vitro digestibility. Similar findings were also reported in finger millet [13], sorghum [28], and napier grass [50]. Remarkably, we identified three accessions (IE 3821, IE 3723, and IE 5435) that had high stover yield, IVOMD, and ME, and low ADF, NDF, and ADL content that can create more palatability and better intake. Forage quality traits improvement need to be such that it should not be exploited at the expense of the grain yield. In our study, we found that grain yield had a non-significant association with IVOMD, ME, Dm, NDF, and ADL, while positively associated with stover yield. Similar findings of a non-significant association between forage quality trait and grain yield were reported in rice [17]. It signifies that there is a huge possibility in finger millet accessions for the development of the dual-purpose crop. Interestingly, a non-significant association

between IVOMD and ME with CP was observed in breeding lines, while these associations were significantly negative in landraces and regions (Table S2a,b). This shows that during the breeding program of cultivar development, there was an improvement in the forage quality positive traits such as in vitro digestibility and crude protein. The IVOMD and ME showed a non-significant association with grain yield, a significantly positive association with stover yield, and a significantly negative association with fiber fractions in the entire set as well as in the breeding lines, landraces, and regions. This indicates that finger millet has a huge potential to be utilized as a dual-purpose (grain and forage) crop.

*4.4. Promising Trait-Specific Sources*

The promising top 10 accessions for each trait and 10 accessions for multiple forage quality traits (Tables 7 and 8) were identified. These accessions could be used as parental lines in a hybridization program for breeding dual-purpose cultivars. The top 10 accessions with multiple forage quality traits identified were from both Africa (Kenya and Uganda) and Asia (India), and most of them were landraces (seven landraces and three breeding lines). Clustering of 314 finger millet accessions using both forage quality and related agronomic traits resulted in five major clusters, and the clustering pattern was mainly based on the performance of accessions for food-feed traits (Tables 5 and 6). Cluster 1 had most of the superior performing accessions for forage quality traits. These clustering and phenotypic distance results would be helpful to identify the diverse parents in the hybridization program to breed finger millet dual-purpose cultivars. Also, three accessions namely IE 4701, IE 3134, and IE 4707 were identified with higher grain yield (>3200 kg ha$^{-1}$) and stover yield (>9400 kg ha$^{-1}$). Of these, IE 4707 had the above mean value for forage quality traits namely CP, ME, IVOMD, and Dm, and lower than mean value (desirable) for ADF, NDF, and ADL. Thus, this accession could be effectively.

## 5. Conclusions

Finger millet is a subsistence crop like other millets, well known for the quality of its grain, which has several health benefits. It is one of the important food crops in the semi-arid regions of the world, providing food and nutritional security. The finger millet stover is used as forage to some extent but not admired like forage sorghum and pearl millet, due to a lack of scientific research on the quantity and quality of finger millet crop residues. The identification of forage value in the finger millet stover offers a unique opportunity to ameliorate the availability of forage to smallholder livestock farmers. This research provides a detailed insight on diversity and variability for forage quality traits in the diverse set of finger millet accessions. Our results showed that finger millet crop residues had higher forage quality than rice and wheat while comparable with sorghum and pearl millet. Thus, it can be added to regular feed supplements for livestock like other cereals crop residues. Identified promising lines can be used in breeding program to develop superior high-yielding cultivars with improved forage quality traits, which could significantly contribute to livestock feeding. This study can give ideas and scope to breeders for breeding dual-purpose finger millet cultivars.

**Supplementary Materials:** The following are available online at https://www.mdpi.com/article/10.3390/agronomy11091706/s1, Table S1: Passport data of 314 finger millet accessions used in the study. Table S2a: Correlation coefficients among forage quality and agronomic traits of breeding lines (*n* = 50; Lower diagonal) and landraces (*n* = 264; upper diagonal) of finger millet germplasm. Table S2b: Correlation coefficients among forage quality and agronomic traits of finger millet accessions from Africa (*n* = 160; lower diagonal) and Asia (*n* = 136; upper diagonal).

**Author Contributions:** M.V. to the conception and design of the study. This work is part of C.B.'s (Chinnadurai Backiyalakshmi) Ph.D. thesis research. C.B. (Chakrapani Babu) and M.V. supported student research as chairman, and co-chairman, respectively. C.B. (Chinnadurai Backiyalakshmi) and M.V. performed the statistical analysis and wrote the first draft of the manuscript. V.P. and K.V.S.V.P. supported forage quality lab analysis. D.N.R. supported phenotypic data collection

and analysis. V.C.R.A. for overall support for this study. All authors have read and agreed to the published version of the manuscript.

**Funding:** This study was undertaken as part of the CGIAR Genebank platform coordinated by Crop Trust, and the CGIAR Research Program on Grain Legumes and Dryland Cereals.

**Institutional Review Board Statement:** Not applicable.

**Informed Consent Statement:** Not applicable.

**Data Availability Statement:** All the required data provided as Supplementary Tables. Researchers can approach the corresponding author for any additional information required.

**Acknowledgments:** The authors are thankful to Ramesh Reddy, Shanku Reddy, Raju Jadhav, and Manish Reddy for their help in field trials; also duly acknowledged ILRI livestock nutritional laboratory, ICRISAT, Hyderabad for their support in the forage trait analysis.

**Conflicts of Interest:** The authors declare no conflict of interest.

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
