# Peer review of "Assessing Forage Potential of the Global Collection of Finger Millet (Eleusine coracana (L.) Gaertn.) Germplasm Conserved at the ICRISAT Genebank"

_agronomy, doi:10.3390/agronomy11091706_

Round 1

Reviewer 1 Report

Overall this is a well planned and  presented study that provides a large amount of information that will be of high value for future finger millet improvement efforts.  Your study design was easy to understand as was your presentation of the data and results.  Minor grammatical revision is suggested (see below) to enhance the readability of the manuscript.  In addition, I would also suggest considering including the R squared value for your NIRS calibration equations (mentioned in line 113, section 2.4) that were utilized for forage quality analysis.  

Grammatical suggestions:

Line 25-27: This sentence is difficult to understand.  I would suggest revising it and including a comma before and after "except for crude protein".

Line 42-44: The sentences in these lines become repetitive to the reader and could possibly be combined into one statement.

Line 45-51: Again, the sentences in these lines are repetitive and all begin in the same way. 

Line 97: Consider exchanging "such as" for "were" since you are listing the 7 agronomic traits.

Line 170: This sentence reads as though a word is missing after "pooled".  Perhaps "pooled values" could be substituted?

Line 454-455: Do you mean all semi-arid regions or specifically where this study was conducted?  This sentence is also difficult to read, please revise to clarify.  Perhaps change to "...region, and ensures food and national security." 

Author Response

We are pleased to submit the revised version of the manuscript ID agronomy-1277795 entitled, “Assessing forage quality of the global collection of finger millet (Eleusine coracana (L.) Gaertn.) conserved at the ICRISAT Genebank” authored by C. Backiyalakshmi, Mani Vetriventhan et. al, after considering reviewers’ suggestions and comments, and the edits were made as track change in the revised version.

"This manuscript is a re-submission of agronomy-1277795".

The detailed response to the reviewers’ comments/suggestions are below.

Reviewers’ comments and suggestions

Reviewer 1

Overall, this is a well-planned and presented a study that provides a large amount of information that will be of high value for future finger millet improvement efforts.  Your study design was easy to understand as was your presentation of the data and results. 

Response: We thank the reviewer for his/her comments and suggestions.

Minor grammatical revision is suggested (see below) to enhance the readability of the manuscript. 

Response: Thanks for the suggestions. Changes have been made in the revised version with track changes

In addition, I would also suggest considering including the R squared value for your NIRS calibration equations (mentioned in line 113, section 2.4) that were utilized for forage quality analysis.  

Response: We have included the R2 value for the NIRS calibration equation in the revised version (line 133-135).

Grammatical suggestions:

Line 25-27: This sentence is difficult to understand.  I would suggest revising it and including a comma before and after "except for crude protein".

Response: Thanks for the suggestions. The sentence has been rewritten in the revised version. (Page: 1, line: 27-29)

Line 42-44: The sentences in these lines become repetitive to the reader and could possibly be combined into one statement.

Response: Thanks for the suggestions. In the revised version, the sentence has been rewritten. (Page: 2, line: 48-50)

Line 45-51: Again, the sentences in these lines are repetitive and all begin in the same way. 

Response: Thanks for the suggestions. The sentence has been rewritten in the revised version. (Page: 2, line: 52-58)

Line 97: Consider exchanging "such as" for "were" since you are listing the 7 agronomic traits.

Response: Thanks for the suggestions. The word “namely” has been added. (Page: 3, line:112)

Line 170: This sentence reads as though a word is missing after "pooled".  Perhaps "pooled values" could be substituted?

Response: Thanks for the suggestions. We have replaced the “pooled” as “pooled data”. (Page: 6, line: 186)

Line 454-455: Do you mean all semi-arid regions or specifically where this study was conducted?  This sentence is also difficult to read, please revise to clarify.  Perhaps change to "...region, and ensures food and national security." 

Response: Thanks for the suggestions. The sentence has been rewritten in the revised version. (Page: 15, line: 477- 478)

Reviewer 2 Report

The authors have done a lot of interesting work. A have few suggestion that are indicated to the attached document. The main is related with the use of term forage, fodder, stover. It is better to be use always the same. Otherwise it is confusing for the reader.

Author Response

Reviewer 2

Comments and Suggestions

The authors have done a lot of interesting work. A have few suggestion that are indicated to the attached document. The main is related with the use of term forage, fodder, stover. It is better to be use always the same. Otherwise it is confusing for the reader.

Response: Thanks to the review the comments and suggestions to strengthen our manuscript. As suggested by the reviewer 2 in the PDF version, the edits have been made in the revised manuscript with track change. The major changes include:

  1. All fodder and forage terms were changes into “Forage” except stover yield.
  2. We have added maturity stage as phenological stage. (Page: 1, line: 24; page: 3, line:121)
  3. We have added the four controls information. (Page: 2, line: 87-88)
  4. We have included the ploidy information about finger millet in “Introduction chapter”. (Page: 1, line: 42-44)
  5. The remaining minor changes suggested by reviewer 2 are also in revised manuscripts with track changes.